# Life Cycle Assessment and Economic Analysis of Biomass Energy Technology in China: A Brief Review

**Shuangyin Chen [1,2,*], He Feng [1], Jun Zheng [2], Jianguo Ye [2], Yi Song [3], Haiping Yang [4] and Ming Zhou [1,2,*]**

[1] China-EU Institute for Clean and Renewable Energy, Huazhong University of Science and Technology, Wuhan 430074, China; m201871270@hust.edu.cn
[2] Institute of New Energy, Wuhan 430206, China; Zheun@sina.com (J.Z.); jye@inew.cn (J.Y.)
[3] The Chinese University of Hong Kong, Shenzhen 518172, China; yisong@link.cuhk.edu.cn
[4] State Key Laboratory of Coal Combustion, Huazhong University of Science and Technology, Wuhan 430074, China; yhping2002@163.com
* Correspondence: sychen@hust.edu.cn (S.C.); zhouming@hust.edu.cn (M.Z.)

**Abstract:** This study describes the technological processes and characteristics of biomass direct combustion power generation, biomass gasification power generation, biomass mixed combustion power generation, and biomass biogas power generation in terms of their importance and application in China. Under the perspective of environmental and economic sustainability, the life cycle assessment (LCA) method and dynamic analysis method based on time value are used to simulate and evaluate the environmental loads and economic benefits of different power generation processes. By comparing with coal-fired power generation systems, the environmental and economic benefits of different biomass power generation technologies are illustrated. The results shows that biomass gasification power generation has the best environmental benefits, with a total load of $1.05 \times 10^{-5}$, followed by biomass biogas power generation ($9.21 \times 10^{-5}$), biomass direct combustion power generation ($1.23 \times 10^{-4}$), and biomass mixed combustion power generation ($3.88 \times 10^{-4}$). Compared with the environmental load of coal-fired power generation, the reduction rate was 97.69%, 79.69%, 72.87%, and 14.56% respectively. According to the analysis of the technical economy evaluation results, when the dynamic pay-back period and IRR (internal rate of return) were used as evaluation indicators, the biomass direct combustion power generation has the best pay-back period (7.71 years) and IRR (19.16%), followed by the biogas power generation, with higher dynamic payback period (12.03 years), and lower IRR (13.49%). For gasification power generation and mixed-combustion power generation, their dynamic payback period is long, and the IRR is low. If net present value (NPV) is selected as the evaluation index, the biogas power generation appears to be the best because its net present value per megawatt is 11.94 million yuan, followed by direct combustion power generation (6.09 million yuan), and the net present value of mixed-combustion power generation and gasification power generation is relatively low. Compared with coal-fired power generation, direct combustion power generation and biogas power generation present significant economic benefits.

**Keywords:** biomass power generation; life cycle assessment; environment load; economic evaluation

## 1. Introduction

The depletion of fossil fuels and environmental deterioration has become the shackles of social development [1–3]. An energy transition towards green and clean renewable energy is essential for the sustainable development of society. Biomass energy has the characteristics of comprehensive sources, abundant reserves, low emissions, and renewable resources with high application potential [4–6]. In

recent years, it has been widely studied by global experts and scholars and is known as the fourth most abundant energy resource after coal, oil, and natural gas [7–11].

Biomass energy has been applied in almost every sector of modern industry. Due to the difference in resource conditions and environmental requirements, the development policies and R&D priorities of biomass energy in various countries are also different. Researches on biomass energy in foreign countries mainly focus on the gasification, liquefaction, pyrolysis, curing, and direct combustion of biomass energy [12–19], while researches in China mainly focus on biomass power generation [20–23]. Since the beginning of the new century, in order to alleviate the dual pressure of future energy and environment, the state has issued a series of biomass promotion policies to support the rapid development of biomass energy and improve the level as well as application scale of domestic biomass power generation. According to statistics, China's total installed capacity of biomass power generation has reached 14.76 million kW by 2016 [24], ranking second in the world.

There are many kinds of biomass power generation technologies in China, including biomass direct combustion power generation, biomass gasification power generation, biomass mixed-combustion power generation, and biogas power generation. Researches on biomass power generation are extensive, mainly focusing on the modelling, optimization, and process evaluation of a single technology or process, and progress have been achieved [25–33]. However, there is no relevant study on the systematic and comprehensive evaluation of several specific biomass power generation technologies. Therefore, from the perspective of environmental sustainability and the industrial economy, the whole life cycle assessment (LCA) and dynamic analysis method respectively based on the time value are used in this paper to analyze the environmental input–output list and the capital input–output list of significant domestic biomass power generation technologies. The environment load and technical economy index are calculated, and the results are compared with coal-fired power to clarify the environmental and economic benefits of different biomass power generation technologies.

## 2. Current Situation of Major Biomass Power Generation Technologies in China

At present, domestic biomass power generation technologies mainly include biomass direct combustion power generation, gasification power generation, mixed-combustion power generation, and biogas power generation. The specific process and characteristics are listed below.

### 2.1. Biomass Direct Combustion Power Generation

Biomass direct combustion technology generates power by feeding biomass raw material into a steam boiler to produce steam for driving the steam turbine and generator for electricity. Equipment, except the boiler, remains the same as the traditional coal-fired power generation system.

In the 1970s, western developed countries began to realize the importance and necessity of biomass, and were committed to conducting research on its technology, development, and application. By the 1980s, biomass direct combustion power generation was pioneered by the Danish government and developed by BWE (Burmeister & Wain Energy). In 1988, the first biomass direct combustion power plant was built. Subsequently, it gradually radiated to the whole world and was listed as one of the principal promotion projects of the United Nations [34].

At present, domestic biomass direct combustion power generation is relatively mature, but problems still exist, both technically and systematically. On the one hand, limited by biomass output, collection radius, combustion temperature, and other factors, the unit capacity is small. Moreover, low operating efficiency and single combustion of biomass raw materials can easily cause severe slagging and corrosion problems. On the other hand, with the large-scale development of biomass direct combustion, policies, incentives, and evaluation mechanisms need to be consistent with the development of technology as well as the industry, which requires further improvement.

Consequently, domestic scientists have carried out plenty of research. Zhang [35] conducted a comprehensive analysis and comparison of raw material collection methods, and power generation methods of direct combustion power generation globally, and clarified its existing problems. Wang [36]

reviewed the current status of global biomass direct-fired power generation, and compared and evaluated the level of biomass direct-fired power generation in China, based on which proposed suggestions for future development directions. On the basis of the research on straw raw material pretreatment technology, Xia et al. [37] analyzed the main factors for improving the efficiency of power generation process, and obtained optimal parameter conditions. By sorting out the research status of biomass raw material pretreatment globally, Ling [38] obtained the best treatment methods for different biomass raw materials as well as the best parameter conditions. By reforming the structure of the boiler superheater, He [39] prevented the slagging problem of the superheater and improved the reliability of the boiler operation. Yang and Jiang [40,41] independently developed the water-cooled vibrating grate, the major equipment of straw direct combustion power generation. According to the analysis of the characteristics of the ash slag in power plant, Mei [42] concluded that the main factors causing slag formation and low thermal efficiency of the boiler were alkali metals, and explored the migration mechanism of alkali metals. Based on the domestic biomass power generation industry, Li [43] analyzed the supporting policies and incentive measures to improve the level of China's biomass direct combustion power generation. Wang [44] analyzed the effect of energy saving, emission reduction and logistics cost of biomass direct combustion power generation projects. Liu et al. [45] linked biomass direct-fired power generation with CDM (clean development mechanism) project and accessed the value of environmental benefits. Duan [46] used LCA to evaluate the risk of biomass direct combustion project.

### 2.2. Biomass Gasification Power Generation

Biomass gasification power generation refers to the gas fuel generated by incomplete oxidation of biomass in a gasifier, which is purified to remove impurities such as tar and then burned in a gas turbine for power generation. This technology has several advantages, such as coal saving, low biomass raw material consumption, high comprehensive power generation efficiency, low environmental emissions, and no slagging and corrosion phenomenon, making it a relatively ideal biomass energy utilization technology. However, since the gas from gasification contains certain impurities, including ash, coke and tar, etc., the impurities need to be removed through the purification system to ensure the normal operation of gasification and power generation equipment. Limited by the key technology of domestic gasification power generation, it is still in the demonstration and research stage.

Researches on biomass gasification power generation mainly focus on gas power generation, gas purification, tar cracking and the optimization as well as benefit evaluation of demonstration project design [47]. Han et al. [48]. expounded the mechanism of tar generation in gasification process, the factors affecting the formation of tar and the removal methods of tar based on the current situation of gas purification globally. Li [49] analyzed the influence factors and removal methods of tar by means of simulation experiment. Li [50] et al. focused on the catalytic cracking in tar cracking and clarified the catalytic effects of different catalysts. Wei [51] used experiments to elucidate the main factors and parameters influencing the catalytic cracking efficiency of tar. Wu et al. [52] introduced the design characteristics of the 4 MW biomass gasification integrated combined cycle power generation demonstration project and explained the optimization technology that could be utilized in the process. Based on the actual operation of the 4 MW gasification power plant in Zhejiang Province, Chen et al. [53] analyzed the main factors affecting the power generation efficiency of the power plant and illustrated the main problems existing in the system.

### 2.3. Biomass Mixed-Combustion Power Generation

Biomass mixed-combustion power generation is a technology that combines biomass and coal to generate electricity, which shows the synergistic effect between coal and biomass fuel. Currently, it has been widely used in foreign countries and there are more than 300 mixed-combustion power plants in the world. It can be achieved by properly modifying the boiler of existing coal-fired power plants. The advantages of biomass mixed-combustion power generation include small investment,

short construction cycle, significant economic benefit, and so on. However, the addition of biomass will increase the difficulty of the pretreatment of raw materials, reduce the theoretical combustion temperature of boilers, cause ash accumulation and corrosion, etc., and have higher requirements on biomass fuel treatment system and combustion equipment [54,55].

Li et al. [56] investigated the combustion ash characteristics of biomass mixed with coal through experiments, and clarified the major influencing factors and the best parameters. Cao [57] established the boiler combustion model based on BP (back propagation) neural network, which helps to achieve the simulation and optimization of the combustion situation of the power plant. Wang et al. [58] analyzed the impact of different biomass on the combustion characteristics of pulverized coal, and concluded that straw can improve the combustion characteristics of pulverized coal. Sun [59] systematically studied the comprehensive combustion characteristics of raw materials and the influence of additives on the comprehensive combustion characteristics of biomass mixed coal, then determined the most dopant ratio of biomass mixed coal. Yuan [60] established a biomass and coal mixed combustion power generation model via mathematical method, clarified major factors and parameters of mixed combustion power generation, and conducted empirical research.

### 2.4. Biomass Biogas Power Genzeration

Biomass biogas power generation is a novel type of power generation technology that integrates energy-saving and environmental protection. Organic and municipal waste are used to produce biogas through fermentation, which is burned later to drive the generating units to generate electricity. This technology is of decisive significance for improving the regional environment, protecting the ecological environment and developing a circular economy by achieving clean energy recycling and reducing environmental emissions. In recent years, there have been many studies on biogas power generation [61,62], including biogas raw material treatment technology, biogas combustion power generation, biogas fuel cell power generation, process optimization, and benefit evaluation, etc. Cong [63] et al., evaluated the significance of biological resource utilization of papermaking sludge in terms of environment and economic. Gao [64] discussed the characteristics and feasibility of anaerobic fermentation of straw through biogas fermentation experiment. Yue [65] et al. analyzed the future development direction of biogas combustion power generation based on biogas combustion characteristics and biogas engine transformation. Zeng [66] et al., implied the application prospect and direction of biogas fuel cell technology in China through comparison with foreign biogas power generation.

## 3. Environmental Benefit Analysis of Domestic Biomass Power Generation Technologies

### 3.1. Goal and Scope

There are many researches on environmental impact assessment of biomass power generation in China, which are generally based on the LCA (life cycle assessment) method [67–70]. Therefore, in order to ensure the accuracy and reliability of the analysis, the LCA method that follows the ISO 14,040 standard is applied in this paper to analyze the full life cycle environmental assessment of biomass power generation, which is then compared with coal-fired power generation in traditional coal-fired power plants. The environmental emission status and emission reduction benefits of various biomass power generation technologies and coal-fired power generation technologies are illustrated.

The process of the LCA method can be divided into four stages: the determination of goal and scope, life cycle inventory analysis, environmental impact assessment, and result interpretation. Due to the complexity of the system that involves several biomass power generation technologies and coal-fired power generation technology, the following simplification and assumptions are made when establishing the evaluation model: the scope of simplified environmental assessment includes the operation stages of raw material production, raw material processing, raw material transportation,

and power generation, while the stages of equipment manufacturing, recovery, depreciation, and plant construction are ignored.

In this study, the case studies of biomass direct combustion, gasification, mixed combustion, biogas, and coal-fired power generation are carried out and the unit capacities are selected to be 30 MW, 4 WM, 300 WM, 2 MW, and 1320 MW, respectively. Taking biomass direct combustion power generation capacity selection as an example, in 2017, China added 16 biomass direct combustion power plants, increasing the total capacity to 544.7 MW, with a median of 30 MW. For convenience of comparison, all project life is assumed to be 20 years, and the functional unit (Fu) is set as the generating capacity of 1 kwh.

### 3.2. Inventory Analysis

Given the environmental impact in the operation stage of the power plant is mainly caused by the emission of gas pollutants, the discharge of gaseous pollutants is primarily considered in the life cycle inventory analysis.

Table 1 shows the emission data of the significant pollutants of biomass direct combustion power generation, gasification power generation, mixed-combustion power generation, biogas power generation, and traditional coal-fired power generation 1 kWh life cycle. The emission data of biomass direct combustion power generation, gasification power generation, biogas power generation, and conventional coal-fired power generation was obtained from literature [71–76] and field investigation of power plants. However, limited by domestic biomass mixed-combustion technology and industrial level, it is difficult to obtain the accurate emission data of biomass mixed-combustion power generation. Therefore, a simplified processing method is adopted in this paper to calculate the emission list of biomass mixed-combustion power generation by using partial biomass direct combustion (20%) instead of the traditional coal-fired power generation.

**Table 1.** Life cycle list of different generation technologies, kg/kWh.

| Technology | $CO_2$ | CO | $CH_4$ | $NO_x$ | PM | $SO_2$ |
|---|---|---|---|---|---|---|
| direct combustion | $7.48 \times 10^{-2}$ | $2.50 \times 10^{-4}$ | $5.29 \times 10^{-5}$ | $3.04 \times 10^{-3}$ | $2.24 \times 10^{-4}$ | $3.26 \times 10^{-3}$ |
| gasification | $4.70 \times 10^{-2}$ | $8.30 \times 10^{-5}$ | $4.71 \times 10^{-5}$ | $1.11 \times 10^{-4}$ | $3.37 \times 10^{-4}$ | $2.58 \times 10^{-4}$ |
| mixed-combustion | $8.71 \times 10^{-1}$ | $1.29 \times 10^{-3}$ | $2.09 \times 10^{-3}$ | $5.78 \times 10^{-3}$ | $1.62 \times 10^{-2}$ | $8.60 \times 10^{-3}$ |
| biogas | $5.11 \times 10^{-1}$ | $1.03 \times 10^{-3}$ | $5.37 \times 10^{-5}$ | $8.72 \times 10^{-4}$ | $1.69 \times 10^{-4}$ | $3.18 \times 10^{-3}$ |
| Coal-fired | $1.07$ | $1.55 \times 10^{-3}$ | $2.60 \times 10^{-3}$ | $6.46 \times 10^{-2}$ | $2.02 \times 10^{-2}$ | $9.93 \times 10^{-3}$ |

### 3.3. Environmental Impact Assessment

According to the conceptual framework of LCA impact assessment stage, the life cycle environmental impact assessment model was constructed. The inventory analysis data was interpreted by the intensity of the contribution of each specific environmental exchange to the determined environmental impact type. Then the environmental impact potential value of each type of impact was calculated, standardized and weighted to obtain the total environmental load.

#### 3.3.1. Environmental Impact Potential Value Calculation

In this study, only five midpoint impact categories are taken into consideration: global warming potential (GWP), acidification potential (AP), creation of photochemical ozone potential (POCP), human toxicity potential (HTP), and soot potential (SP). The measured impact potentials of these five impacts are reasonably close to the corresponding real potentials, the evaluation results are relatively accurate and representative. Freshwater and marine aquatic ecotoxicity are omitted from the impact assessment indictors because of inherent uncertainties in fate-exposure-effect modelling of emissions that contribute to freshwater and marine pollution.

Table 2 shows major pollutants and their equivalent factors contained in different environmental impact categories, e.g., GWP is measured in $CO_2$-equivalents. The characterization factor of $CO_2$

emission is 1, the characterization factor of CO emission is 2, the characterization factor of $CH_4$ emission is 25, and the characterization factors of $NO_x$ emissions is 320. The same procedure may be easily adopted to obtain the detailed characterization factors of AP, POCP, HTP, and SP. Combined with the life cycle list of different generation technologies in Table 1, the potential value of product environmental impact is calculated as follows:

$$EP(j) = \sum EP_i(j) = \sum [Q_i \times EF_i(j)] \tag{1}$$

*EP(j)*: Contribution of product environmental impact potential value *j*;

*EP$_i$(j)*: Contribution of the pollutant *i* to the environmental impact *j*;

*Q$_i$*: Emissions of pollutant *i*;

*EF$_i$(j)*: Equivalent factor of the pollutant *i* to the environmental impact category *j*;

and the characterization results of different power generation processes are shown in Table 3.

**Table 2.** Characterization of each impact category.

| Impact Categories | Unit | Key Parameters |
|---|---|---|
| global warming potential (GWP) | kg $CO_2$ eq | $CO_2$ = 1, CO = 2, $CH_4$ = 25, $NO_x$ = 320 [77] |
| acidification potential (AP) | kg $SO_2$ eq | $SO_2$ = 1, $NO_x$ = 0.7 [78,79] |
| creation of photochemical ozone potential (POCP) | kg Ethene eq | $SO_2$ = 0.048, $NO_x$ = 0.028, CO = 0.04, $CH_4$ = 0.007 [79,80] |
| human toxicity potential (HTP) | kg CO eq | $SO_2$ = 100, $NO_x$ = 65, CO = 1 [73,81] |
| soot potential (SP) | kg PM eq | PM = 1 [81] |

**Table 3.** Characterization results of different power generation processes.

| Technologies | GWP/kg $CO_2$ eq | AP/kg $SO_2$ eq | POCP/kg Ethene eq | HTP/kg CO eq | SP/kg PM eq |
|---|---|---|---|---|---|
| direct combustion | 1.05 | $5.39 \times 10^{-3}$ | $2.49 \times 10^{-4}$ | $5.24 \times 10^{-1}$ | $2.24 \times 10^{-4}$ |
| gasification | $8.39 \times 10^{-2}$ | $3.36 \times 10^{-4}$ | $1.80 \times 10^{-5}$ | $3.31 \times 10^{-2}$ | $3.37 \times 10^{-4}$ |
| mixed-combustion | 2.77 | $1.26 \times 10^{-2}$ | $6.24 \times 10^{-4}$ | 1.24 | $1.62 \times 10^{-2}$ |
| biogas | $7.93 \times 10^{-1}$ | $3.79 \times 10^{-3}$ | $2.05 \times 10^{-4}$ | $3.76 \times 10^{-1}$ | $1.69 \times 10^{-4}$ |
| Coal-fired | 3.21 | $1.45 \times 10^{-2}$ | $7.18 \times 10^{-4}$ | 1.41 | $2.02 \times 10^{-2}$ |

### 3.3.2. Normalization and Weighting Analysis

In order to compare the relative values of various environmental impact potentials, different environmental impact categories were standardized. This paper uses the standard person equivalent conceptual model proposed by Yang et al. [73,81], and takes 1990 as the base year for standardization. The results are shown in Table 4.

**Table 4.** Normalized reference value and normalized result.

| Impact | Baseline [72,82] | Normalized Result | | | | |
|---|---|---|---|---|---|---|
| | | Direct Combustion | Gasification | Mixed-Combustion | Biogas | Coal-Fired |
| GWP | 8700 kg $CO_2$ eq | $1.21 \times 10^{-4}$ | $9.64 \times 10^{-6}$ | $3.19 \times 10^{-4}$ | $9.12 \times 10^{-5}$ | $3.68 \times 10^{-4}$ |
| AP | 36 kg $SO_2$ eq | $1.50 \times 10^{-4}$ | $9.32 \times 10^{-6}$ | $3.51 \times 10^{-4}$ | $1.05 \times 10^{-4}$ | $4.01 \times 10^{-4}$ |
| POCP | 0.65 kg Ethene eq | $3.83 \times 10^{-4}$ | $2.78 \times 10^{-5}$ | $9.60 \times 10^{-4}$ | $3.16 \times 10^{-4}$ | $1.10 \times 10^{-3}$ |
| HTP | 9100 kg CO eq | $5.76 \times 10^{-5}$ | $3.63 \times 10^{-6}$ | $1.36 \times 10^{-4}$ | $4.13 \times 10^{-5}$ | $1.55 \times 10^{-4}$ |
| SP | 18 kg PM eq | $1.24 \times 10^{-5}$ | $1.87 \times 10^{-5}$ | $9.00 \times 10^{-4}$ | $9.39 \times 10^{-6}$ | $1.12 \times 10^{-3}$ |

If the normalized result of different environmental impact categories is the same, it does not mean that the potential environmental impact is equally serious. Therefore, it is necessary to rank

the importance of the different impact categories; that is, assign different weights to different impact categories, distinguish their harm degree to the total environmental impact, and then compare them. This process is called weighted assessment.

The weight is determined in reference to the expert ranking and analytic hierarchy process (AHP) in this paper. AHP can effectively solve the complex system with multiple criteria, decompose environmental problems into elements at different levels to form a hierarchical structure, and then compare the elements at each level in combination with expert opinions, determine their importance and value, and establish a judgment matrix A (Table 5).

**Table 5.** Judgment matrix A.

| Weighting | Global Warming | Acidification | Photochemical Ozone Creation | Human Toxicity | Soot |
|---|---|---|---|---|---|
| Global warming | 1 | 3 | 6 | 5 | 4 |
| Acidification | 1/3 | 1 | 3 | 6 | 5 |
| Photochemical ozone creation | 1/6 | 1/3 | 1 | 3 | 6 |
| Human toxicity | 1/5 | 1/6 | 1/3 | 1 | 3 |
| Soot | 1/4 | 1/5 | 1/6 | 1/3 | 1 |

The eigenvector, the largest eigenvalue and the consistency ratio are solved using MATLAB software. The results show that the eigenvector is (0.48, 0.05, 0.13, 0.25, 0.09), the largest eigenvalue is 5.22, the consistency index and ratio are 0.056 and 0.0495, respectively.

Since the consistency ratio (CR) is less than 0.1, the eigenvector can represent the weight coefficient of different environmental impact types. Therefore, the weight coefficients of GWP, AP, POCP, HTP, and SP are 0.48, 025, 0.05, 0.13, and 0.09, respectively, as shown in Table 6.

**Table 6.** Weighting result.

| Weight | Global Warming | Acidification | Photochemical Ozone Creation | Human Toxicity | Soot |
|---|---|---|---|---|---|
| Value | 0.48 | 0.25 | 0.05 | 0.13 | 0.09 |

The total environmental loads of different generation processes are further calculated, and the results are shown in Table 7.

**Table 7.** Total environmental loads.

| Technology | Direct Combustion | Gasification | Mixed-Combustion | Biogas | Coal-Fired |
|---|---|---|---|---|---|
| Value | $1.23 \times 10^{-4}$ | $1.05 \times 10^{-5}$ | $3.88 \times 10^{-4}$ | $9.21 \times 10^{-5}$ | $4.56 \times 10^{-4}$ |

*3.4. Results Analysis*

According to Table 7, biomass gasification power generation has the best environmental benefits ($1.05 \times 10^{-5}$), followed by biomass biogas power generation ($9.21 \times 10^{-5}$), biomass direct combustion power generation ($1.23 \times 10^{-4}$), and mixed-combustion power generation ($3.88 \times 10^{-4}$). Compared with the environmental load of traditional coal-fired power generation, the emission reduction benefits are 97.69%, 79.69%, 72.87%, and 14.56%, respectively.

## 4. Economic Analysis of Domestic Biomass Power Generation Technologies

Based on the economic parameters in the construction and operation periods of the biomass power plant project, the economic benefit calculation model of the biomass power plant was constructed in

order to carry out the economic evaluation of the power plant. The initial investment in the construction period mainly includes civil construction, equipment procurement and installation, equipment commissioning and delivery, etc. This period is considered as the capital outflow stage in this paper. The operation period refers to the period from the formal production to a scrap of the project. The capital outflow stage mainly includes raw materials, fuel procurement and transportation, labor management costs, equipment depreciation, and other operating costs. In contrast, the capital inflow stage mainly includes electricity and other by-product income.

In this paper, the economic analysis of domestic biomass power generation technologies is conducted from three aspects: the analysis of capital inflow and outflow list, the economic evaluation of biomass power generation, and the analysis of the economic evaluation results of biomass power generation technology.

### 4.1. Capital Inflow–Outflow List Analysis

Based on the literature and the actual situation of the power plant during its construction and operation periods, various capital input–output data of representative biomass power plants and coal-fired power plants in China were obtained and then calculated [71,74,75,82–84]. The results are shown in Table 8. During data processing, the following assumptions were made: according to a regulation issued by National Development and Reform Commission in 2006, the annual benchmark rate of return of the biomass power station was set to be 10%, with an operating period of 20 years, an annual generating period of 5500 h, and a feed-in tariff of 0.0941 pound/kWh, etc.

**Table 8.** Capital inflow–outflow list of different power generation technologies, Euro (€).

| Technology | Scale | Initial Investment | Operation Investment | Operation Profit | Annual Profit |
|---|---|---|---|---|---|
| direct combustion | 30 MW | 33,607,200 | 8,878,320 | 15,518,250 | 6,639,930 |
| gasification | 4 WM | 7,255,644 | 1,205,094 | 2,069,100 | 864,006 |
| mixed-combustion | 300 WM | 150,323,250 | 110,918,808 | 116,403,804 | 5,484,996 |
| biogas | 2 MW | 12,079,782 | 407,550 | 2,178,198 | 1,770,648 |
| Coal-fired | 1320 MW | 661,426,062 | 337,885,284 | 425,765,604 | 87,880,320 |

### 4.2. Economic Evaluation

The dynamic analysis method based on time value was used to evaluate the economy of different biomass power generation technologies and coal-fired power generation technology. Discounted pay-back (10%) period, net present value (NPV), and internal rate of return (IRR) are selected as specific evaluation index, and the formula is as follows:

$$\text{Discounted pay} - \text{back } (10\%) \text{ period} : \sum_{t=0}^{P'_t} (CI - CO)_t (1 + i_c)^{-t} = 0 \tag{2}$$

$$\text{NPV} : NPV = \sum_{t=0}^{n} (CI - CO)_t (1 + i_c)^{-t} \tag{3}$$

$$\text{IRR} : IRR = i_1 + (i_2 - i_1) \frac{NPV_1}{NPV_1 + |NPV_2|} \tag{4}$$

$P_t'$: Discounted Pay-back (10%) period
$CI$: Cash inflow
$CO$: Cash outflow
$(CI–CO)_t$: Net cash flow in year $t$
$n$: Years

$i_c$: Base earning rate

*NPV*: Net present value $i_1$: Low discount rate

$i_2$: High discount rate

The dynamic investment pay-back period, NPV, and IRR of different biomass power generation technologies as well as coal-fired power generation technology were calculated using the Formulas (2)–(4) with data from Table 8, and the results are shown in Table 9.

**Table 9.** Economic evaluation results of different power generation technologies.

| Technology | Scale | Dynamic Pay-Back Period/Year | NPV/Pound·MW$^{-1}$ | IRR/% |
|---|---|---|---|---|
| direct combustion | 30 MW | 7.71 | 763,686 | 19.16 |
| gasification | 4 MW | 19.28 | 25,080 | 10.20 |
| mixed-combustion | 300 MW | >20.00 | 155,496 | −2.83 |
| biogas | 2 MW | 12.03 | 1,497,276 | 13.50 |
| Coal-fired | 1320 MW | 14.50 | 566,808 | 11.88 |

Note: the benchmark rate of return of coal-fired power generation is 8%.

*4.3. Results Analysis*

According to the economic evaluation results (Table 9) of different biomass power generation technologies, when the dynamic pay-back period and IRR are used as evaluation indexes, the biomass direct combustion power generation has the best pay-back period (7.71 years) and IRR (19.16%), followed by biomass biogas power generation technology (12.03 years and 13.49%, respectively). The dynamic payback period of biomass gasification and mixed-combustion power generation technologies is longer, and the IRR is lower. If NPV is the evaluation index is used, biomass biogas power generation has the highest NPV (1,497,276 pound/MW), followed by biomass direct combustion power generation technology (763,686 pound/MW), and the NPV of biomass mixed-combustion and gasification power generation are lower. Compared with coal-fired power generation, direct combustion power generation and biogas power generation have more economic benefits.

## 5. Conclusions

(1)  This paper discussed the process flow and characteristics of four leading biomass power generation technologies, including biomass direct combustion power generation, biomass gasification power generation, biomass mixed-combustion power generation, and biomass biogas power generation.

(2)  According to the environmental impact assessment of coal-fired power generation and different biomass power generation technologies, the latter has better environmental benefits, among which biomass gasification power generation has the best environmental benefits, with environmental loads of $1.05 \times 10^{-5}$. Compared with coal-fired power generation, the emission reduction rate of biomass gasification power generation is 97.69%, followed by biomass biogas power generation (79.69%), biomass direct combustion power generation (72.87%), and biomass mixed-combustion power generation (14.56%).

(3)  According to the economic evaluation of different biomass power generation technologies, when the dynamic pay-back period and IRR are used as evaluation indexes, the biomass direct combustion power generation has the best pay-back period (7.71 years) and IRR (19.16%), followed by biomass biogas power generation (12.03 years and 13.49%, respectively). The dynamic pay-back period and IRR of biomass gasification and mixed-combustion power generation technologies are longer and lower, respectively. When taking the NPV as the evaluation index, biomass biogas power generation technology has the highest NPV (1,497,276 pound/MW), followed by biomass direct combustion power generation technology (763,686 pound/MW), and the NPV of biomass

mixed-combustion and gasification power generation are lower. Compared with coal-fired power generation, direct combustion power generation and biogas power generation have more obvious economic benefits.

**Author Contributions:** Conceptualization, S.C. and M.Z.; Methodology, S.C., H.F., J.Z., J.Y., Y.S., H.Y. and M.Z.; Software, S.C. and M.Z.; Validation, S.C., H.F., J.Z., J.Y., Y.S., H.Y. and M.Z.; Formal analysis, S.C., J.Z., J.Y. and M.Z.; Investigation, S.C. and H.F.; Resources, S.C., H.F., J.Z., J.Y., Y.S., H.Y. and M.Z.; Data Curation, S.C., J.Z., J.Y. and M.Z.; Writing-Original Draft Preparation, S.C.; Writing-Review & Editing, S.C. and M.Z.; Visualization, S.C., H.F., J.Z., J.Y., Y.S., H.Y. and M.Z.; Supervision, S.C., J.Z., J.Y., Y.S., H.Y. and M.Z.; Project Administration, S.C., J.Z., J.Y., Y.S., H.Y. and M.Z.; Funding Acquisition, S.C. and M.Z. All authors have read and agreed to the published version of the manuscript.

**Funding:** This research received no external funding.

**Acknowledgments:** This research is supported by the National Key R&D Program of China (NO. 2018YFB1501403), the Foundation of State Key Laboratory of Coal Combustion (No. FSKLCCA1902), and the Double first-class research funding of China-EU Institute for Clean and Renewable Energy (No. 3011120016). We also thank Qing Yang from Huazhong University of Science and Technology for her generous help.

**Conflicts of Interest:** The authors declare no conflict of interest.

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
