# Peer review of "Life Cycle Assessment and Economic Analysis of Biomass Energy Technology in China: A Brief Review"

_processes, doi:10.3390/pr8091112_

Round 1
Reviewer 1 Report
The paper deals with the environmental impact of different biomass technological processes, also coping with the economic issues.
The paper needs to be revised because the authors did not detail the methods used for their calculations, especially dealing with the environmental issues, which are the core of the paper. They should state (not enough referencing to Chinese literature very difficult to find) the hypotheses, the methods and results should be commented on. Not only reporting the numbers in the Tables.
They only detailed the economic part using very basic economic indicators. All the economic values should be reported either in dollars (preferred) or in Euros which are international currencies. Yuan is not an international one.
References are only from Chinese authors, except one which is wrongly referenced. In addition, the cite the authors with Name and Surname, which is very confusing. In the References various styles are employed and should be leveled out.
Please see the following details:
Line 89: The reference to authors (Zhang Peiyuan [35] )is miselading. Do the authors use Name and Surname? If so, please use only surname. IN this case I think it is Peiyuan.
Line 195: "The functional unit is set to be 10000 kWh-"What is the time span? Using this low value for energy production means a very small unit.
Table 1: Listed data require units of measure. I.e. CO2 is in kg/kWh or what else? The same with the other components.
Line 218 and 219: It is very hard to understand what the authors want to explain in this sentence. It should be rewritten with a clearer statement.
Line 224: The authors should reference the values they present here. For example, for GWP do the authors employed data from IPCC? Which year? Because the values for CH4 and NOx are a little different from the usual values (IPCC XXXX)
Line 226: "the results are shown in table 3"- The auhtors should detail which is the procedure they employed to obtain such results. It's not sufficient to state that they used ISO 14040. Operational procedures should be detailed, togheter with the boundaries of the systems.
Line 233: "Yang Jianxin et al."- references need to be revised since the authors are referenced with name and surname (very confusing) and in this case the ref number [72-76] is wrong becasue they refer to multiple different authors.
Line 239: "The obtained nine-scale..."- from where? Who assessed this Judgment MAtrix?
Table 5: the authors should explain why used these weights in evaluating the environmental impact
Line 242-244:the authors assume that everyone should know the method they employed. Since it is difficult to obtain the reference, I think they should detail the employed method
Line 280: "etc."-not enough "etc." for example what is the interst rate assumed? How did you evaluate the "Annual profit"? Before or after taxes? Is the revenue from selling electric energy only due to feed-in-tariff?
Table 8: the economic analysis should be carried out using an international currency, such as Dollar $ or Euro €
Table 9: "NPV/10 thousand yuan/ MW"- yuan should be changed to Dollars or Euro
References: Many references are wrongly formatted. They should be formatted following the Paper rules. In addition, it seems that only Chinese researchers study these topics. Authors should consider also authors from different countries.
ref. 2: What is this? "Luca" is a name and could not be the surname of an author. Also it is not clear which kind of paper is this: journal, report, Mater thesis.
ref. 14: different and wrong formatting
ref. 16:different and wrong formatting
Author Response
Dear Reviewer:
Thank you for your comments concerning our manuscript entitled “Anaerobic Digestion Processes” (ID: 860720). Those comments are all valuable and very helpful for revising and improving our paper, as well as the important guiding significance to our researches. We have studied comments carefully and have made correction which we hope meet with approval. Revised portion are marked in red in the paper. The main corrections in the paper and the responds to the reviewer’s comments are as flowing:
Responds to the reviewer’s comments:
1.
Line 89: The reference to authors (Zhang Peiyuan [35] )is misleading. Do the authors use Name and Surname? If so, please use only surname. IN this case I think it is Peiyuan.
Response1:According to the opinions of reviewer, the author references are uniformly sorted out, and the surname is uniformly changed.
Line 195: "The functional unit is set to be 10000 kWh-"What is the time span? Using this low value for energy production means a very small unit.
Response2: Thanks for your suggestions. We admit that it is not appropriate when missing the time span. We have adjusted accordingly and added a time span, also we have written some explain, which has been marked in red.
Table 1: Listed data require units of measure. I.e. CO2 is in kg/kWh or what else? The same with the other components.
Response3: Thanks for your suggestions. We admit that it is not appropriate when listing data without units of measure. We have added a reasonable unit and you could find a explain above, which has been marked in red.
Line 218 and 219: It is very hard to understand what the authors want to explain in this sentence. It should be rewritten with a clearer statement.
Response4: Thanks for your suggestions. We admit that there are some problems in this sentence, which may confuse you when reading it. We have rewritten this sentence whit a clearer statement and marked it in red in the text.
5.
Line 224: The authors should reference the values they present here. For example, for GWP do the authors employed data from IPCC? Which year? Because the values for CH4 and NOx are a little different from the usual values (IPCC XXXX)
Response 5: Thanks for your suggestions. References have been supplemented in this paper.
6.
Line 226: "the results are shown in table 3"- The authors should detail which is the procedure they employed to obtain such results. It's not sufficient to state that they used ISO 14040. Operational procedures should be detailed, together with the boundaries of the systems.
Response 6: Thanks for your suggestions. The calculation process has been added in this paper.
7.
Line 233: "Yang Jianxin et al."- references need to be revised since the authors are referenced with name and surname (very confusing) and in this case the ref number [72-76] is wrong because they refer to multiple different authors.
Response7: Thanks for your suggestions. We admit that it is not appropriate to use both the name and surname and we did make a mistake when writing the ref number. We have deleted the name of the author and rewritten the ref number.
8-10.
Line 239: "The obtained nine-scale..."- from where? Who assessed this Judgment MAtrix?
Table 5: the authors should explain why used these weights in evaluating the environmental impact.
Line 242-244: the authors assume that everyone should know the method they employed. Since it is difficult to obtain the reference, I think they should detail the employed method
Response 8-10: Thanks for your suggestions. The above opinions have been modified according to the opinions of reviewers and marked in red in this paper.
11.
Line 280: "etc."-not enough "etc." for example what is the interest rate assumed? How did you evaluate the "Annual profit"? Before or after taxes? Is the revenue from selling electric energy only due to feed-in-tariff?
Response 11: Thanks for your suggestions. The assumed data are from national standard documents, which have been marked red in this paper.
12.
Table 8: the economic analysis should be carried out using an international currency, such as Dollar $ or Euro €
Response 12: Thanks for your suggestions. We admit that it is not appropriate to use a Chinese currency when it comes to the economic analysis. We have adjusted accordingly to convert the original unit to an international unit (€), and also the corresponding figures both in the Table 8 and the text have been converted.
Table 9: "NPV/10 thousand yuan/ MW"- yuan should be changed to Dollars or Euro
Response 13: Thanks for your suggestions. We acknowledge that it is improper to use a Chinese unit here. We have adjusted accordingly to convert the original unit to an international unit (€), and also the corresponding figures both in the Table 9 and the text have been converted.
14-17.
References: Many references are wrongly formatted. They should be formatted following the Paper rules. In addition, it seems that only Chinese researchers study these topics. Authors should consider also authors from different countries.
ref. 2: What is this? "Luca" is a name and could not be the surname of an author. Also it is not clear which kind of paper is this: journal, report, Mater thesis.
ref. 14: different and wrong formatting
ref. 16: different and wrong formatting
Response 14-17: Thanks for your suggestions. All reference formats have been revised.

Reviewer 2 Report
Dear Authors,
your work is good, a little rewording is required.
Hope you would be able todo it.
Good luck.
Author Response
Dear Reviewer:
Thank you for your comments concerning our manuscript entitled “Anaerobic Digestion Processes” (ID: 860720). Those comments are all valuable and very helpful for revising and improving our paper, as well as the important guiding significance to our researches. We have studied comments carefully and have made correction which we hope meet with approval. Revised portion are marked in red in the paper.
Thank you very much !

Round 2
Reviewer 1 Report
Some major revisions are required for considering publication of the paper.
Line 163: This sentence should be removed. Such plants are widley spread and employed around the world since the end of XX century, mainly in Europe, but also in USA and some other countries.
Line 170: I do not understand why authors persist to find only Chinese references (in this case not clear what kind of document it is) instead of looking for international references from well known authors.
Line 195: Numbers should be seprated from the unit of measure. The authors should justify the use of these values for the capacities of different plants. Is it because these are the average capacities in China for such plants?
Line 196: Not clear why Fu ihas this value: 10000 kWh (with capital W) or 10 MWh? For the environmental impact it would be right also considering 1 kWh. So pollution could be given in g (or kg)/kWh
Line 220: The authors should indicate the acronyms (GWP, AP, etc.) since they are used in the next without any explanation
Line 231: Maybe the index "i" should be written after EP:
Epi(j)
Table 3: Please, specify that the results are evaluated based on the Fu (=10000 kWh)
Table 4:Baseline: The authors should motivate this number in a very convincing way.
Line 243: Maybe the sentence should read as:
"If the normalized results of different ... " otherwise the sentence is useless.
Table 5: The authors should clearly state why these are the weights of each environmental impact. The sentence above is not a clear explanation for a scientific paper.
Line 295: The economic evaluation is useless without knowing which is the interest rate the authors used.
I suppose that annual interst rate is 10%, but the authors should clearly state it.
Equation (2): I would prefer the authors use: Discounted Pay-back period instead of "Dynamic" which is used hardly ever.
Equation (4): Not clear to me why the authors use this formula. From where? Which is the reference?
I know that IRR is the interest rate that vanishes the NPV after n years. Using formula (4) needs two NPVs and two interest rates. Which are in this case?
References:
Some references are wrong or not clear. For esample ref.14 what does it mean Conference paper??? Which conference, where? Use the appropriate format. But this reference is not the only one wrong. Please address all the references to give a clear indication of the paper
References 76-79:
These references need to be written following the format required. They are badly written and formatted
Author Response
Dear editor and reviewer,
Thank you very much for your valuable opinions. We have revised the paper according to your opinions.
The main modifications are as follows:
To reviewer:
1.
Line 163: This sentence should be removed. Such plants are widely spread and employed around the world since the end of XX century, mainly in Europe, but also in USA and some other countries.
Thank you very much for your comments. This sentence has been deleted in this paper.
2.
Line 170: I do not understand why authors persist to find only Chinese references (in this case not clear what kind of document it is) instead of looking for international references from well known authors.
As foreign experts have little research literature on Biomass power generation technology in China, this paper mainly cites Chinese domestic literature.
3.
Line 195: Numbers should be separated from the unit of measure. The authors should justify the use of these values for the capacities of different plants. Is it because these are the average capacities in China for such plants?
Thank you for your suggestion, and the modification has been completed according to your suggestion.
4.
Line 196: Not clear why Fu I has this value: 10000 kWh (with capital W) or 10 MWh? For the environmental impact it would be right also considering 1 kWh. So pollution could be given in g (or kg)/kWh
It has been modified according to your opinion, and pollution emissions (kg) at 1kWh have been taken into account.
5.
Line 220: The authors should indicate the acronyms (GWP, AP, etc.) since they are used in the next without any explanation
Thank you for your suggestion. The abbreviation has been modified
6.
Line 231: Maybe the index "i" should be written after EP:
Epi(j)
Thank you very much.
7.
Table 3: Please, specify that the results are evaluated based on the Fu (=10000 kWh)
Thank you for your suggestion, and calculate the characteristic result according to formula 1 in the paper.
8.
Table 4: Baseline: The authors should motivate this number in a very convincing way.
The baseline data are from references 72 and 82, which have been noted in this paper.
9.
Line 243: Maybe the sentence should read as:
"If the normalized results of different ... " otherwise the sentence is useless.
It has been modified in accordance with your opinion.
10.
Table 5: The authors should clearly state why these are the weights of each environmental impact. The sentence above is not a clear explanation for a scientific paper.
In this paper, we mainly use the method of expert ranking to determine the importance of impact types, and then determine the weight according to the analytic hierarchy process.
11.
Line 295: The economic evaluation is useless without knowing which is the interest rate the authors used.
I suppose that annual interst rate is 10%, but the authors should clearly state it.
It has been modified according to your opinions. Thank you
12.
Equation (2): I would prefer the authors use: Discounted Pay-back period instead of "Dynamic" which is used hardly ever.
It has been modified according to your opinions. Thank you
13.
Equation (4): Not clear to me why the authors use this formula. From where? Which is the reference?
I know that IRR is the interest rate that vanishes the NPV after n years. Using formula (4) needs two NPVs and two interest rates. Which are in this case?
Interpolation method is used to calculate the IRR. Take biogas power generation as an example, the specific process is as follows:
When, i1=0.13,NPV1=358580.3
When, i2=0.14,NPV2=-352549
So, =0.13+(0.14-0.13)×[358580.3/(358580.3+352549)]=0.135
References:
Some references are wrong or not clear. For esample ref.14 what does it mean Conference paper??? Which conference, where? Use the appropriate format. But this reference is not the only one wrong. Please address all the references to give a clear indication of the paper
Thank you very much for your instruction and help. We have revised and verified the references
15.
References 76-79:
These references need to be written following the format required. They are badly written and formatted
Thank you very much for your instruction and help. We have revised and verified the references